# Determining the Effects of Pelleted Cranberry Vine Grains on the Ewe and Offspring during Pregnancy and Lactation

**DOI:** 10.3390/ani13121989

**Published:** 2023-06-14

**Authors:** Delaney Smith, Katherine Petersson, Maria L. Peterson

**Affiliations:** Department of Fisheries, Animal, and Veterinary Sciences, University of Rhode Island, Kingston, RI 02881, USA; delaneysmith@uri.edu (D.S.); kpetersson@uri.edu (K.P.)

**Keywords:** periparturient, anthelmintic, ovine, milk fat, condensed tannins

## Abstract

**Simple Summary:**

Parasitic infections are a significant problem in the sheep and goat industries worldwide. This issue is compounded by parasites becoming resistant to commercially available dewormers. Currently, the efficacy of tannins and other plant secondary compounds is being evaluated as natural dewormers in sheep and goats. The main objective of this study was to determine if cranberry vine was safe to feed pregnant and lactating ewes. The results of this study indicate that there were minimal effects on the growth and health of the ewe and her offspring from the consumption of a 50% cranberry vine pellet during late gestation and lactation.

**Abstract:**

When creating any new anti-parasitic interventions, it is important to evaluate their effects across all life stages. This study had three objectives, which were to evaluate the effect of feeding cranberry vine pellet (CVP) on (1) ewes’ body weights and BCS during late gestation and lactation; (2) ewes’ milk quality during lactation; and (3) lambs’ body weight and growth parameters from birth to 65 days of age. Across two years, 41 Dorset ewes were fed either a 50% CVP or a matching control pellet (CON) from 104 ± 1.60 days of gestation for 62.8 ± 0.68 days of lactation. Measurements were collected from ewes (BW, BCS, and milk) and lambs (BW and body size). Milk from CVP ewes exhibited reduced milk fat and solids (*p* < 0.01) and increased concentrations of milk urea nitrogen (*p* = 0.02) when evaluated for the treatment–time. There was no significant difference in the BCS, protein, lamb BW, or growth measurements for treatment–time (*p* ≥ 0.05). Additional research that targets blood biochemistry and metabolic assessments is needed to fully determine the impact of this pellet on ewes and lambs.

## 1. Introduction

Drug resistance among gastrointestinal nematodes (GIN) is a significant problem in the sheep industry [1]. For example, 15% of all sheep deaths in the US in 2019 were caused by parasitic infections [2]. Furthermore, there are stages during production where animals become more vulnerable to parasite infections [3]. One of these stages, known as the periparturient period (PPR), is associated with a rise in fecal egg counts [4,5] during the last few weeks of a ewe’s gestation and can continue into lactation. This is because the ewes’ body allocates more energy to growing the fetus and making milk over immune function [6,7,8]. Compounding this issue is the ever-increasing population of drug-resistant GIN, which is the result of deworming protocols that have traditionally been used for generations [9]. This has culminated in a significant reduction in the efficacy of commercially available dewormers [10,11]. Therefore, there is a critical need to find alternative methods to control GIN. Plants, specifically those that contain polyphenols such as condensed tannins (CTs), are being evaluated as a natural alternative to chemical dewormers. The goal is to use these tannins alone or in conjunction with chemical dewormers [12,13,14,15]. 3Specifically, plants such as wormwood (*Artemisia absinthium* [16,17]), mallow (*Malva sylvestris* L. [17]), *Sericea lespedeza* [18,19], birds-foot trefoil (*Lotus corniculatus* [14,20,21]), and cranberry vine (*Vaccinium macrocarpon*, [22]) are presently being evaluated using in vitro and in vivo methods. The goal is that using these plants will be able to provide a sustainable option for parasite control for producers in both conventional and organic markets.

Our research group has determined that using cranberry vine powder in vitro has anthelmintic effects on parasites, specifically the gastrointestinal helminth *Haemonchus contortus* [22]. The mechanism of action is postulated to be the CTs and other plant secondary compounds that are contained in the vine [23]. Overall, the presence of CTs can increase antioxidant capacity [24,25], which in turn helps reduce the risk of other diseases [26,27]. Additionally, plants that are high in CTs have been shown to bind to proteins and create more bioavailable molecules that are then easily used by the rumen [28,29,30]. Therefore, feeding CTs could potentially have other positive effects on livestock that could potentially benefit their health and production.

While the effects of feeding CTs to animals during PPR have potential, a significant amount of care needs to be taken to ensure that this can be conducted safely. As mentioned previously, the effects of CTs are not limited to the parasite and could potentially impact the animals’ metabolism as well as their overall production ability. The PPR encompasses a time when ewes are pregnant and then lactating, which is a critical time for these ewes, both metabolically and production-wise. Therefore, understanding the potential changes in the composition of the milk (fat, protein, total solids, and nitrogen concentrations) is important, as this not only nourishes the offspring but is relevant for dairy production [31]. Variations in lactation have been observed as direct effects of CTs and polyphenol supplementation. Studies have shown that diets high in CTs have varying effects on milk fat [32,33], milk protein [34,35], and milk urea nitrogen concentrations [36,37] across a number of ruminant species when fed during lactation. If there is any impact of the CTs on the ewe’s milk quality or composition, it is key to elucidate this information before recommending this feed type to dairy producers.

When feeding supplements, additives, or providing treatments during gestation and lactation, one must also consider the potential effects that this could have on the resulting offspring. Producers rely on the revenue generated from selling the lambs produced by the ewes or use female lambs as replacements in the flock [2,38]. It is well established that changes to the maternal diet during pregnancy can have both positive and adverse effects on the growth and development of the offspring and their subsequent productivity later in life [39,40]. For example, maternal under- and over-nutrition during pregnancy has been found to negatively impact offspring muscle growth and development as well as carcass adiposity as they mature [39,40,41]. Some studies have demonstrated that diets containing variations in feeds, such as changing protein amounts [42], restricting or overfeeding the rations [40,43], and the inclusion of non-traditional grains such as dried distillers grain (sorghum) [44], soybean hulls [45], starch-treated wheat [46], and even hydroponic barley sprouts [47], can affect the growth capacity of the lambs through various life stages.

However, there have been no studies that have evaluated the effect of feeding cranberry vine grains during the periparturient period on the ewe, her milk composition, or her offspring. Therefore, this is a significant gap in knowledge that must be filled before feeding cranberry vine grains during the PPR can be recommended to producers.

The main objective of this study was to determine if cranberry vine was safe to feed pregnant and lactating ewes. In this study, the effects of supplementation with a 50% cranberry vine pellet (CVP) on the production of the ewe and her offspring were determined. Understanding the effects of maternal consumption of a 50% CVP on the dam and her offspring is essential before recommending this product as a treatment for preventing or limiting the effects of nematode infection during the periparturient period. We hypothesize that the inclusion of a 50% CVP into the diet of periparturient ewes from late gestation through 65 days of lactation will (1) not impact ewe weight gain or loss versus control ewes; (2) have minimal effects on milk composition versus control ewes; and (3) lambs born to CVP ewes will exhibit similar ADG to CON-born lambs.

## 2. Materials and Methods

### 2.1. Cranberry Vine Pellet

This study took place over two years: spring of 2020 (YR1) and spring of 2021 (YR2). Cranberry vine (Stevens variety, *Vaccinium macrocarpon*) was grown and harvested by A.D Makepeace (Wareham, MA, USA) and transported to the University of Rhode Island (URI, Kingston, RI, USA) Agronomy farm in fall of 2019. The vines were stored in a greenhouse in rows to allow them to dry. Vine piles were flipped twice daily for 2 weeks to ensure even drying and to prevent molding. After the vine was sufficiently dry, all vines were ground into a powder using a tractor (M4 Series, Kubota, Grapevine, TX, USA) and a hammer mill attachment (MMRB20 Bison Hammer Mill, ½ inch screen, Aguascalientes, México). The cranberry vine powder was incorporated into a 50% CVP by Green Mountain Feeds (Bethel, VT, USA) formulated to be equivalent in digestible dry matter (DDM) to the control (CON) feed (18% protein, Kalmbach Feeds, Sandusky, OH, USA). Additional molybdenum was included in the formulation to counteract the increased concentrations of copper naturally found in the vine. This run of the CVP was fed for both years of the study and reanalyzed prior to use in YR2. All forage analyses for these studies were conducted by the Forage Laboratory at Dairy One Co-op Inc. (Ithaca, NY, USA). During YR2 of the study, the composition of the CON pellet changed due to supply and production issues due to COVID-19. During YR2, the CON pellet contained 16% crude protein (16% protein, Kalmbach Feeds, Sandusky, OH, USA).

### 2.2. Animals

Purebred Dorset ewes and their offspring from the URI Peckham Farm (Kingston, RI, USA) were used for these studies. A power analysis using an α of 0.05 and a power (1-β) was performed prior to the start of the study to ensure that we had a sufficient number of ewes per each treatment group. Each year, we had more ewes per treatment group than was determined by our power analysis.

### 2.3. Ewe Classification and Treatment Groups

This project was conducted in collaboration with another researcher who was evaluating the effect of CVP consumption on parasitic infections during pregnancy and lactation (i.e., the periparturient period and data pending publication). This portion of the study was designed to determine the effect of CVP consumption during the periparturient period on production parameters for ewes and their offspring. Dorset ewes (YR1: *n* = 25; YR2: *n* = 16) were pasture-bred by a Dorset ram. The ram wore a marking harness, and an ewe was considered bred the day that she was marked by the ram. These dates were used to determine the ewe’s approximate due date. Ewes were ultra-sounded to confirm pregnancy, viability of the pregnancy, and approximate number of lambs 60 days after introduction of the ram. All efforts were taken to balance the ewes evenly across treatment groups for the following factors: average fecal egg count, age, weight, predicted number of offspring, and parity. Eight weeks prior to the first expected due date, study ewes were separated into two treatment groups. Ewes were housed in group pens, with one treatment per pen, as follows: CVP (YR 1 *n* = 12, YR 2 = 10) or CON (YR 1 *n* = 13, YR 2 = 10). Ewes were fed at or above the National Research Council (NRC [48]) requirements for a ewe expecting twins, depending on their gestational stage. During this time, ewes were fed first-cutting hay and measured amounts of the grain created and described in Section 2.1 based on their assigned treatment group. Grain was weighed out for the entire group based on the NRC standards mentioned above, and they were fed twice a day throughout the trial in large feed bunks that were present in both treatment pens. Postpartum and throughout lactation, ewes were receiving second-cutting hay instead of first cutting in addition to their grain. All feed amounts were adjusted according to the NRC requirements at this stage for lactating ewes that were nursing twins. In YR1, ewes began being fed the CON or CVP diet on day 108 ± 1.18 of gestation, and they remained on the diets until day 62 ± 1.18 postpartum and YR2 started the diet on day 100 ± 2.01 of gestation and remained on the diets until day 68 ± 2.01 postpartum. In total, ewes were on treatment diets for 119 ± 1.71 and 115 ± 2.10 days for YR 1 and 2, respectively. Ewes were treated as individuals throughout the trail, and the results were analyzed according to the number of ewes, not the number of treatment pens. All variables were standardized so that the week that the ewe gave birth was considered their week 0 timepoint. All values taken during pregnancy were designated as a negative week value (i.e., the week number before birth), and all values taken during lactation were designated as a positive week value (i.e., the week number after birth). All ewes were provided free-choice access to water and supplementary minerals for the duration of the study.

### 2.4. Feed Sample Collection

Throughout both years of the trial, feed and hay subsamples were routinely collected and stored at −20 °C for a final composite analysis. Following the conclusion of each trial, stored composite samples were thoroughly homogenized and sent off for analysis. Composite feed samples from the study were analyzed (Table 1) by the Forage Laboratory at Dairy One Co-op Inc. (Ithaca, NY, USA).

### 2.5. Ewe Sample Collection

For both year 1 and 2 of the study, weekly body weights (BWs) and body condition3 scores (BCS) were taken to monitor ewes’ health during pregnancy and lactation. Body condition scores were evaluated for each ewe using the methods outlined in Greiner, 2012 [49]. Adjustments were made to the amount of feed ewes received based on increases in their BW or changes in the production stage (i.e., lactation).

### 2.6. Milk Collection and Analyses

Milk samples were collected weekly after lambing by hand. Ewes were restrained, teats were cleaned with water, and fore was stripped prior to collection. Twenty milliliters of milk, split as evenly as possible between each half, was collected in a sample tube containing a preservative. Samples were shipped to Dairy One Co-op Inc. (Ithaca, NY, USA) for the analysis of milk components, including fat, protein, total solids, and milk urea nitrogen concentrations.

### 2.7. Lamb Body Weight and Growth Measurements

Body weights and measurements (crown–rump length, girth, and height) of each lamb were collected within the first 24 h after birth and repeated at weaning (week 8). All measurements were performed using a seamstress’ measuring tape. Crown–rump was recorded as the length of the lamb, starting between their ears, to the base of their neck and then ending at the base of their tail. Girth measurements were taken around the ribcage, along the heart girth line, with the measuring tape gently placed around the body. The height measurement was recorded while the lamb was slightly suspended above the ground, causing their legs to dangle freely and not bend. The measurement was taken from between the shoulder blades to the flat bottom of the hoof. The average daily gain (ADG) was determined using their birth weight and weaning weights.

### 2.8. Statistical Analyses

Data were analyzed in SAS using PROC MIXED with repeated measures (time) and Kenward–Rogers. Ewes’ BWs were split into two different timepoints for analyses: pregnancy and lactation. Covariance structures were selected based on the lowest AIC value. Covariance structures used were Huynh–Feldt (milk protein and solids), Unstructured (MUN), Toeplitz (Ewe BW lactation), Autoregressive (Ewe pregnancy BW), and Compound Symmetry (milk fat). Data were considered significantly different when *p* ≤ 0.05. Paired T-tests were also performed on lamb measurements as well as ewe and lamb average daily gains (ADGs).

## 3. Results

### 3.1. Feed Analysis

Nutrient analyses were similar between study years, with minor variations in protein and TDN (Table 1). Specifically, the protein concentration in the CVP feed was 1.6% less in YR 1 (17.7%) than YR 2 (19.3%; Table 1). The grain TDN, which was used for calculating feed rations, was 2% less in YR1 (54%), than in YR2 (52%; Table 1). Through our research, we determined that cranberry vine, while rich in CTs and polyphenols, also contains greater copper concentrations (34 ppm). The copper concentrations in the CVP were 35 ppm and 37 ppm in YR 1 and 2, respectively (Table 1).

### 3.2. Ewes’ Body Weight

No effect was observed for the treatment–timepoint on the ewes’ body weight during pregnancy (*p* = 1.00) or lactation (*p* = 0.99) during either year of the project. As anticipated, the ewes’ body weights steadily increased over time across treatments prepartum during each trial year (YR1 *p* = 0.0002; YR2 *p* < 0.0001). During pregnancy in YR1, CON ewes weighed more across all timepoints (including the start of the study) than CVP ewes (*p* = 0.0002, Figure 1A). This was not the case in YR2 (*p* = 0.27, Figure 1C), when ewes’ BWs were the same to begin with. During lactation, there was a 0.68% increase in YR1 and a 9.92% reduction in YR2 in the weight of ewes across all timepoints (Figure 1B,D). Therefore, it can be stated that the groups were similar in BW at the beginning of the trial and remained that way throughout. Any variations in the group starting weights can be attributed to differences in the expected number and actual number of lambs that a ewe was pregnant with for the two groups (Table 2).

There were no differences over time across treatments (treatment–time interaction) during lactation in YR1 (*p* = 0.61) or YR2 (*p* = 0.56). However, there was an effect of treatment during lactation on the ewes’ body weight. During the lactation period, CVP ewes on average weighed 80.47 kg ± 3.56 kg and 78.56 kg ± 2.41 kg in YR1 and YR2, respectively, whereas CON ewes weighed 90.08 kg ± 3.19 kg and 84.11 kg ± 4.21 kg (*p* ≤ 0.01, Figure 1B,D). It is important to note that in YR1, the CON ewes did weigh more to begin with and throughout the duration of the study. However, in YR2, the weights were similar between these two groups of ewes to begin with, and the CVP ewes exhibited a modest 6.9% difference in BW during the lactation period when compared with CON ewes. Therefore, these data suggest that the consumption of CVP during lactation may have an impact on ewes’ BW.

Furthermore, body condition scores were monitored throughout both trials, with an average range of 2.86 ± 0.07 to 3.01 ± 0.04 during gestation and 2.88 ± 0.05 to 3.13 ± 0.04 during lactation for both treatments in both years. No differences in ewe BCS were observed (treatment–time, time, or treatment) when evaluated within each trial (*p* ≥ 0.64). Therefore, the changes that we observed in BW were in the absence of changes in BCS. Body condition scores are a subjective measurement, and for future studies, especially in animals being reared for meat production, measuring back-fat thickness via ultrasound may yield some more precise data.

### 3.3. Ewe Milk Parameters

#### 3.3.1. Milk Fat

We determined that cranberry vine consumption during lactation did appear to have a marginal effect on ewe milk fat. Specifically, an effect of the treatment–timepoint for the milk fat percentage in YR1 of this study was observed (*p* < 0.01). One-week postpartum, the CVP treatment group had greater concentrations of milk fat (*p* < 0.01; Figure 2A). However, by week two, the CVP group’s percentage of milk fat had dropped considerably, while the CON groups dramatically increased (*p* < 0.01; Figure 2A). From week 3 through weaning (week 8), there was no difference in the milk fact concentrations between groups. In YR 2 (2021) of this trial, no effect of the treatment–time was observed on the milk fat concentrations in CVP ewes when compared to CON. However, an effect of treatment was observed with CON ewes having a greater milk fat concentration than CVP ewes overall (*p* = 0.009; CON 7.69 ± 0.19, CVP 6.65 ± 0.23; Figure 2B). It should be noted that the milk fat concentration in both groups in YR2 were, on average, greater than the milk fat concentrations measured in YR1.

#### 3.3.2. Milk Protein

No effect of treatment–timepoint was observed in YR1 or YR2 (*p* > 0.05). However, there was an observed effect of the timepoint for the milk protein percentages in YR1 (2020) of this study (*p* < 0.01; Figure 3A). As can be observed in Figure 3A, the protein concentrations for both treatments decreased throughout the postpartum period similarly (Figure 3A). Specifically, CVP ewes exhibited a 24% decrease in milk protein from week 1 to week 8, and CON ewes had a 19% decrease. No effect of the timepoint was observed in YR2 (*p* > 0.05). In YR2 (2021) of this trial, there was an overall effect of treatment (*p* = 0.03; CON 4.51 ± 0.04, CVP 4.36 ± 0.04; Figure 3B). While this effect is statistically significant, the reduction in milk protein concentrations is relatively small.

#### 3.3.3. Total Solids (TS)

There was an effect of the treatment–timepoint in YR1 (2020) of this study (*p* < 0.01; Figure 4A). One-week postpartum, the CVP treatment group demonstrated significantly greater concentrations of total solids (TS) than the CON group (*p* < 0.01; Figure 4A). However, by week two, the CVP group’s ppm of solids dropped considerably (*p* < 0.01; Figure 4A). From week 3 and beyond, no difference in TS was observed between CON and CVP ewes (*p* ≤ 0.07). This follows the same pattern of the milk fat observations for YR1 and is likely the cause of this difference in TS.

In YR2 of this trial, no effect of the treatment–timepoint was observed on TS between CON and CVP ewes. However, there was an effect of treatment (*p* < 0.01), with CON ewes exhibiting greater concentrations of TS (CON 18.23 ± 0.16) than the CVP group (CVP 16.90 ± 0.23; Figure 4B).

#### 3.3.4. Milk Urea Nitrogen (MUN)

Milk urea nitrogen concentrations were significantly affected by the CVP in both years of the trial. In YR1 (2020), no effect of the treatment–timepoint was observed (*p* = 0.18). However, overall, there was an effect of treatment on MUN concentrations (*p* < 0.01: Figure 5A). Specifically, the CVP treatment group had significantly greater MUN concentrations (31.19 mg/dL; Figure 5A) than the CON treatment group (28.75 mg/dL; Figure 5A).

Similar effects were observed in YR2, where MUN was affected over time by the treatments provided to ewes (*p* = 0.03; Figure 5B). CVP ewes exhibited increased MUN concentrations at weeks 2, 3, 4, 5, and 8 postpartum (*p* ≤ 0.03; Figure 5B) when compared with CON animals. Over time, MUN concentrations increased by 14% in CVP animals, whereas CON animals exhibited a 7% reduction in MUN during the lactation period (*p* = 0.04; Figure 5B).

### 3.4. Lamb Body Measurements

There was no observed significance of the treatment–timepoint for any of the lamb body measurements (BW, CRL, height, and girth; *p* ≥ 0.18). All the values did increase over the course of the study from birth until weaning for both years of the trial (*p* < 0.01), resulting in an effect of time. There was no effect of treatment on lamb BW, CRL, height, or girth (*p* ≥ 0.06, Table 3). This was expected and normal, as lambs grow rapidly during this time period [50].

In YR1, between birth and weaning, the CVP group had an ADG of 0.39 kg/day, and the CON group had an ADG of 0.43 kg/day (*p* < 0.001). In YR2, between birth and weaning, the CVP group had an ADG of 0.33 kg/day, and the CON group had an ADG of 0.32 kg/day in BW (kg; *p* > 0.05). Lambs in both treatment groups exhibited similar growth rates throughout both years of this trial. There was no effect of just treatment when separated by year (2020 *p* = 0.07; 2021 *p* = 0.77).

## 4. Discussion

The results from this study are novel, as there has not been any in vivo research evaluating the effects of cranberry vine on periparturient ewes. Additionally, this trial has been one of the few to utilize the vine and stems of a plant whose fruit is well known for its medicinal properties [23,51,52,53]. Based on our data, we have discovered some interesting and relevant caveats that need to be considered when using cranberry vine during pregnancy and lactation.

With regard to our feeds, we ensured that the grain was safe to feed to sheep. Upon milling the cranberry vines, we determined that they had concentrations of copper that can be problematic for sheep. It is well understood that high concentrations of copper (20 ppm or above; [54]) can be toxic to sheep. However, increasing feed supplementation with sodium molybdate is effective at negating the toxic effects of copper [55]. Specifically, it is recommended that for every 3–10 ppm of copper, 1 ppm of molybdenum be fed to prevent copper toxicity [55]. Knowing this, increased amounts of sodium molybdate were added during the creation of the experimental CVP pellet that was fed during the trial.

Our formulated pellet had an initial ratio for Cu:Mo of 12.5 ppm Cu:1 ppm Mo. The researchers acknowledge that this is greater than the aforementioned recommendation. However, additional molybdenum was provided to these animals via other feedstuffs in the diet. The first-cutting hay that was fed had a Cu:Mo of 1.15:1, and the second-cutting hay that was fed had a Cu:Mo of 0.99:1. All animals also had access to free-choice minerals, which also contained trace amounts of sodium molybdate. It is important to note that there were no documented cases of copper toxicity between the 2 years. Furthermore, a preliminary feeding trial of the CVP used in this study was performed with a group of 7-month-old Dorset lambs prior to the start of this trial in the fall of 2019 (data in preparation). Liver samples were taken from those lambs after being on the CVP or CON diet for 6 weeks, and hepatic copper concentrations were measured. On average, hepatic copper concentrations in the CVP group were greater than the CON group (231.25 ug/g vs. 65.08 ug/g; *p* = 0.003). However, copper concentrations need to be at or above 400 ug/g in the liver for toxicity to occur [56].

As we are among the first to try feeding trials using cranberry vines, this information is important to include to help create guidelines for future studies involving sheep or other less copper-sensitive species (i.e., goats and cattle). Furthermore, with the information learned, we have reformulated our pellet for current studies to create the ideal Cu:Mo in the pelleted feed.

### Ewes’ Body Weight

When feeding animals any new diet, it is critical to ensure that it does not negatively impact the BW or growth of the animals. Therefore, we monitored the ewes for changes in BW and BCS over the duration of the study. We observed a reduction in CVP ewes’ BW during the YR1 and YR2 studies that are like those in Mkhize et al. [57], where sheep fed CTs had lower ADGs than CON animals. It has been postulated that the reduction in ADG, and in our case, BW, could be the result of the feed composition in combination with the CTs. For example, an insufficient corn and/or soybean to tannin ratio or the addition of sugary substances (i.e., polyethylene glycol or molasses) can negatively impact the way the body metabolizes food due to the inclusion of CTs [58,59]. To make the cranberry grain more palatable, we did have to add greater amounts of molasses to the pellet as well as remove a lot of the corn filler to include the powdered cranberry vine, so it is possible that this could have affected these results. Additionally, more highly controlled studies need to be performed to determine if this effect on BW is consistent and what some of the metabolic effects/causes are.

Milk production is a key aspect of a ewe’s ability to rear her lambs. Therefore, it was imperative for us to evaluate the effects of the CVP grain on the ewes’ milk composition. It is important to note that our milk fat, protein, and total solid concentrations are similar in both treatment groups to those observed in other studies evaluating sheep milk composition. There are 40 known breeds of dairy sheep and 180 breeds of sheep that are considered to be of dual purpose worldwide [60]. While differences in breed have not been shown to affect milk composition [61,62,63], differences in feeds have been known to contribute to the variations observed in milk components from what is considered standard [64,65,66]. For example, studies have shown that while there is a correlation between the type of feed and the animal’s fatty acid profile, there is no correlation between breed and fatty acid makeup within ovines [61,62,63,64,65,66,67,68,69].

Our findings provide key information that can be important for producers interested in using CVP or other tanniferous forages in dairy animals, where changes in milk composition can impact profitability. Therefore, while our milk fat concentrations were still within the normal range, we did observe differences between the diets.

Feeding CVP to the ewes did result in a modest reduction in milk fat concentrations. As to be expected, differences in the milk total solids were observed as well. Our findings are similar to that in the literature, where the TS tend to follow similar trendlines as the results of the milk fat analyses [30,32,33,35,36,37,70]. Fluctuations in milk fat concentrations can occur due to an animal’s age, stage of lactation, health, and nutritional inputs [61,62,63,64,65,66,67]. This is demonstrated by the reduction in milk fat over time that was observed in our ewes from birth to weaning, which is to be expected [63,71]. Additionally, all the ewes used in the study were similar in age as well as BCS, and the data were standardized to the date that each ewe gave birth. Therefore, these variables cannot be considered factors. The amount of hay/fiber consumed can also impact milk fat [72,73]. Specifically, milk protein concentrations will be reduced, but milk fat concentrations will be greater in animals that are fed diets that are higher in fiber [72]. We did not measure the amount of hay consumed by these ewes in Y1 and Y2, which is a source of variation that we have been able to control in subsequent studies. Additionally, minimal differences in milk protein concentrations were observed at the aforementioned timepoints. Therefore, it is plausible that differences in the fiber/energy content of the feeds may not be to blame.

It Is notable that changes in milk fat concentrations have been observed when feeding CTs to lactating ruminants. Our findings are similar to those of other studies that used multiparous dairy goats and fed high concentrations of CT in early lactation [37,74,75]. In contrast, other researchers using cows and dairy sheep have determined that supplementing with CTs resulted in increased milk fat concentrations [24,26,32]. Our study has a more comprehensive, longitudinal analysis of milk composition than the aforementioned study performed in sheep [32], which only measured sheep milk composition at day 50 post-lambing. Therefore, it can be stated that the effects of CT can vary between species and studies, depending on the design.

As mentioned at the beginning of our discussion section, our milk fat concentrations are similar to what is observed in other studies using sheep and can be considered normal [60,67,76]. While we did observe a reduction in milk fat in CVP animals, the BW and growth of the lambs did not appear to be significantly affected.

However, for animals being used for dairy production, the fluctuations in milk fat in CVP animals may need to be considered, as changes to milk fat composition can impact the amount of money earned from fluid milk as well as its usage in secondary products.

While we did observe a reduction in milk protein as an effect of treatment, it is important to note that this difference is small. These results are consistent with other studies that have found that milk protein concentrations remain stable when animals are supplemented with CTs. The decrease over time in protein concentrations that we observed was similar between both groups, so it was likely some other factor that affected this measurement. Some studies have shown that secondary polyphenolic compounds can occasionally decrease the efficacy of proteolytic enzymes, which can cause decreases in milk protein percentages [24,69,77,78,79,80]. Additional work needs to be performed to better understand why this is occurring. Currently, we are evaluating ewe blood samples and metabolomics to determine if the CVP fed during lactation is impacting these animals metabolically, and thus contributing to some of the modest changes in milk fat and protein.

One of the milk components that did appear to be significantly impacted by feeding the CVP was MUN. Normal MUN concentrations range from 12.2 to 25.8 for sheep on high-energy rations and 12.9–26.7 for sheep on low-energy rations [81]. Our CON ewes for YR2 fell within this range, and YR1 was just outside the upper limit. However, the CVP ewes were significantly greater than these limits. In dairy cattle, MUN can be affected by a variety of different factors, including breed, seasonality, time of sampling, as well as nutrition [82]. Samples were taken from ewes at similar times of the day, and all animals were the same breed and gave birth during the same season, thus eliminating these potential sources of variation. Feeding excessive amounts of protein can also contribute to greater concentrations of MUN in milk. While protein concentrations in the CVP group feed were greater in YR2, this was not the case in YR1; therefore, this may not be a contributing factor. This could, however, explain why the MUN concentrations did appear to be greater in the CON animals in YR1 than they were in YR2.

Our findings are contradictory to what has been reported in the literature, where MUN concentrations are reduced among CT treatment groups in multiple species, including dairy cows, dairy goats, dairy sheep, and water buffaloes [30,33,34,36,37]. Traditionally, high concentrations of MUN are linked to increased nitrogen recycling and protein degradation within the rumen [36,37]. The protein-binding capacity is heavily influenced by the structure of the CTs, which can then have direct impacts on the nitrogen output in an animal’s urine, feces, and milk [34]. For future studies, it would be beneficial to quantify the amount and type of CTs in the pellet in order to have a more detailed understanding of the effects that these CTs have. Additionally, more work needs to be performed to understand what is occurring with these animals metabolically, especially with their energy balance, as this could be impacting the MUN concentrations. Furthermore, studies need to be performed to evaluate the effects of feeding a CVP ration during various life stages on rumen pH, rumen microflora, and the subsequent effects on rumen protein utilization.

Despite changes to the ewes’ milk composition, the lambs continued to grow well. Lambs from the CVP ewes did appear to have a very small reduction in ADG in Y1, but this was not observed in Y2. Furthermore, this reduction was so small that we are not sure that it bears any relevance production-wise. Any observed effect of the timepoint in this study on lambs’ BW or growth measurements can be attributed to the normal growth of these lambs as they increased in size over time [50]. It is extremely difficult to link the effects of tannins to any growth parameters [83], especially when the animals we were observing were not directly fed any of the CVP itself. What we can conclude, however, is that the CVP seemed to have little to no obvious outward maternal programming effect on the lambs. We cannot state that the CVP did not have any long-term effects or impact on organ development, tissue development, or metabolism without subsequent analyses of blood samples or additional studies. This is an area that needs to be explored further, as well as evaluating the concentrations of the CTs in the milk of ewes. This would allow us to determine if the lambs are ingesting any of the CTs while nursing from the dam.

## 5. Conclusions

This study helps contribute to the development of safe alternatives to transitional anthelmintics. Feeding CVP grains to ewes did appear to have modest effects on milk fat, total solids, and protein. CVP animals also exhibited greater concentrations of MUN. There was no effect on the ewes’ body weight or any of the lambs’ measurements. The findings in this paper suggest that feeding cranberry vine during the periparturient period will not have any impact on the body condition of the ewes or the health and growth capacity of the lambs. However, careful consideration should be taken when formulating these rations for lactating animals. Future investigations into tannin concentrations, biomarkers of oxidative stress, blood metabolic changes, and bioactive compounds in essential organs would all be beneficial for a more in-depth understanding of the whole-body effects of tannin-rich compounds on periparturient animals.

## Figures and Tables

**Figure 1 animals-13-01989-f001:**
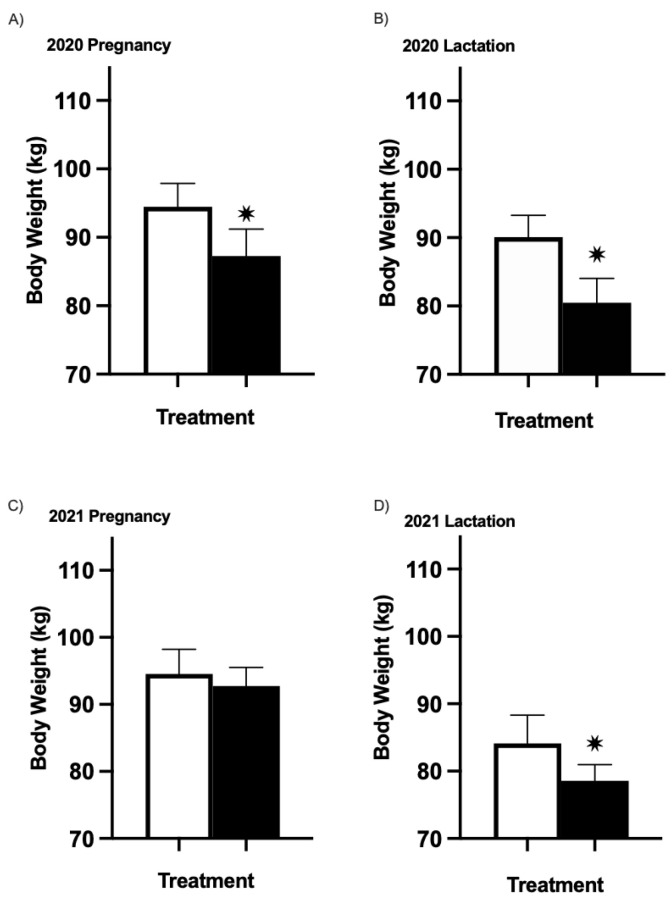
Comparative body weights of study ewes for both years. All untreated CON groups are represented in white, and treatment CVP groups are in black. Body weights for ewes CON (*n* = 13) or treated with CVP (*n* = 12) during pregnancy (**A**) and lactation (**B**) in YR1. Body weights for ewes, either CON (*n* = 8) or treated with CVP (*n* = 8), during pregnancy (**C**) and lactation (**D**) in YR2. Asterisk (*) represents *p* values < 0.001 relative to control on each figure. Error bars represent SEM.

**Figure 2 animals-13-01989-f002:**
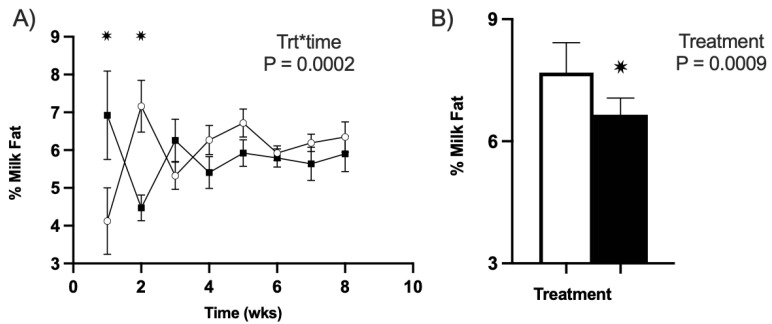
Milk fat percentages of study ewes for both years. All untreated CON groups are represented in white, and treatment CVP groups are in black. (**A**) Milk fat percentage within treatment over time for YR1 (* *p* < 0.005 versus CON within week). (**B**) Milk fat percentage between treatments for YR2 (* *p* = 0.0009 versus CON).

**Figure 3 animals-13-01989-f003:**
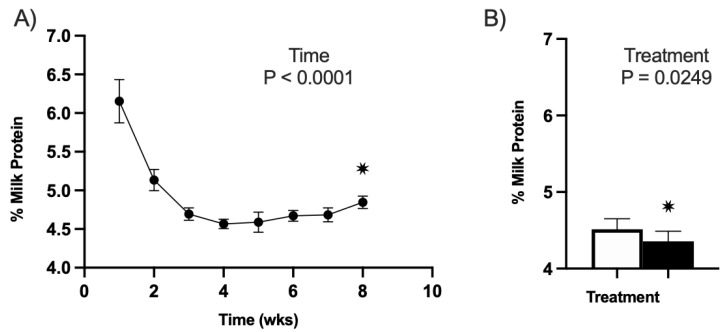
Milk protein percentages of all ewes used in trial. All untreated CON groups are represented in white, and treatment CVP groups are in black. (**A**) Milk protein percentage for all ewes over time for YR1. There was a significance of timepoint of *p* < 0.01. The one line in black does not indicate treatment but is all values averaged together. (**B**) Milk protein percentage between treatments for YR2. The effect of treatment was significant with a * *p* < 0.05 relative to the control group.

**Figure 4 animals-13-01989-f004:**
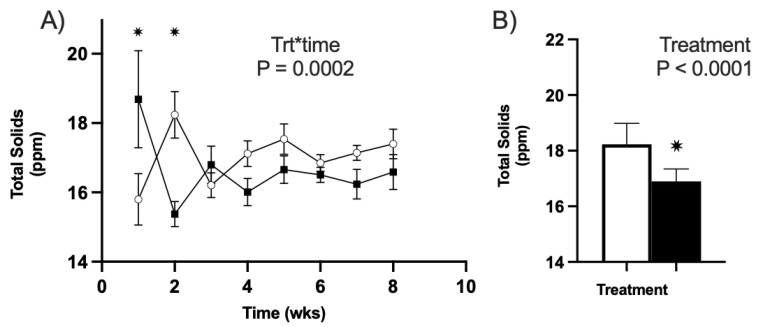
Milk total solids (TS) of all ewes used in trial. All untreated CON groups are represented in white, and treatment CVP groups are in black. (**A**) Milk TS over time for YR1. There was a significance of treatment–time of * *p* < 0.01 with individual significance at weeks 1 and 2 after birth, which are marked with an asterisk. (**B**) Milk TS between treatments for YR2. The effect of treatment was significant with a * *p* < 0.01 relative to CON.

**Figure 5 animals-13-01989-f005:**
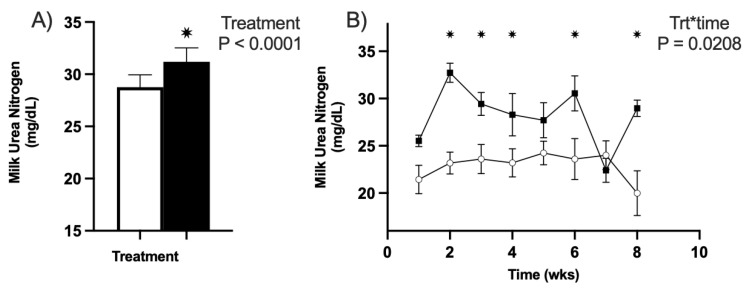
Milk urea nitrogen concentrations of all ewes used in trial. All untreated CON groups are represented in white, and treatment CVP groups are in black. (**A**) Milk urea nitrogen amounts between treatments for YR1. The effect of treatment was significant with a * *p* < 0.01 relative to the control group. (**B**) Milk urea nitrogen over time for YR2. There was a significance of treatment–time of * *p* < 0.05 with individual significance at weeks 2, 3, 4, 6, and 8 after birth, which are all marked with an asterisk. *p* values for individual weeks were all < 0.05.

**Table 1 animals-13-01989-t001:** Feed Nutrient Analysis for Trial Grains.

Feed Type	CP ^1^	DM ^2^	TDN ^3^	Copper (ppm ^4^)	Molybdenum (ppm)
CVP ^5^					
YR1	17.7	90.7	54	35	2.8
YR2	19.3	88.7	52	37	3.1
CON ^6^					
YR1	18.7	86.5	62	11	10.9
YR2	15.8	87.8	62	11	11.4
Hay					
First Cut	10.3	93.2	49	6	5.2
Second Cut	13.1	84.4	51	7	7.1

^1^ CP = crude protein percentage; ^2^ DM = dry matter percentage; ^3^ TDN = total digestible nutrients; ^4^ PPM = parts per million; ^5^ CVP = cranberry vine pellet; ^6^ CON = control pellet.

**Table 2 animals-13-01989-t002:** Ewe parity distribution and number of lambs born.

Treatment	Number of Ewes	Primiparous	Multiparous	Number of Lambs	Singles	Twins	Triplets
2020 Total	25	6	19	44	8	36	0
CVP	12	3	9	21	5	16	0
CON	13	3	10	23	3	20	0
2021 Total	16	5	11	28	6	16	6
CVP	8	2	6	14	3	8	3
CON	8	3	5	14	3	8	3

**Table 3 animals-13-01989-t003:** Distribution of averages for lamb body measurements across both trial years ^7^.

	2020					2021				
	Birth		Weaning		*p*-Value	Birth		Weaning		*p*-Value
Treatment	CON	CVP	CON	CVP		CON	CVP	CON	CVP	
Body weight (kgs)	5.04 ± 0.29	4.99 ± 0.54	27.29 ± 2.02	24.88 ± 1.97	0.48	5.26 ± 0.54	5.22 ± 0.44	21.74 ± 3.79	22.45 ± 1.90	0.53
Crown–rump length (cm)	45.04 ± 0.62	46.55 ± 0.70	79.83 ± 1.16	80.78 ± 1.36	0.14	45.54 ± 1.06	45.38 ± 1.24	75.88 ± 1.62	78.67 ± 1.76	0.53
Girth (cm)	44.28 ± 0.54	44.78 ± 0.89	77.67 ± 1.51	77.98 ± 1.10	0.41	42.98 ± 0.74	43.50 ± 1.20	76.08 ± 1.88	78.50 ± 0.96	0.36
Height (cm)	41.96 ± 0.60	40.80 ± 0.89	69.37 ± 0.77	68.45 ± 0.88	0.07	46.17 ± 0.89	44.67 ± 0.49	63.27 ± 1.42	64.29 ± 1.25	0.88

^7^ *p*-values are for a *t*-test between treatments. Weaning is designated to 8 weeks after the lamb was born. Significance is deemed 0.05 or lower.

## Data Availability

The data presented in this study are available on request from the corresponding author. The data are not publicly available at the request of our collaborators.

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
