# Peer review of "Determining the Effects of Pelleted Cranberry Vine Grains on the Ewe and Offspring during Pregnancy and Lactation"

_animals, 2023, doi:10.3390/ani13121989_

Round 1

Reviewer 1 Report

do da tradução

    Authors, congratulations for yet another work on the subject. Looking at the authors' results, I can identify publication potential, but when I read important sections of the paper, I cannot identify this research importance. Here are some comments, which deserve to be reviewed by researchers: 1) In the introduction section, the authors do not make it clear what is the knowledge gap that exists. What are the questions that previous work by the group or other researchers cradled this research. 2) in the introduction section, there is a hypothesis of what might happen; but there is no TECHNICAL JUSTIFICATION explaining why the use of Cranberry Vine could change body weight, metabolic variables or the quality of milk in particular. 3) the authors' aim is "supplementation". I ask the authors if this term is correct in this study? to supplement is to give something that the animals already consume in a conventional and traditional diet. Is Cranberry Vine used in a traditional sheep diet? or is it a different additive introduced into the diet? did the animals in the control group consume Cranberry Vine in their basal diet? I'm saying all this to encourage researchers to question whether the term supplementation is correct in this study. 5) more details about the pre-experiment sheep is important; about their dates of birth, whether it was in the same week, whether they occurred over time and how the groups were formed. what these animals consumed. All this information on methodology helps the reviewer to understand the study and verify that there was no external interference. 6) the results found had a superficial discussion, which can be deepened. 7) in the conclusion section, there is no conclusion, as there is a summary description of the results. In the conclusion section we must use relative terminology, which allows conclusions from measured variables. The authors described what happened to "the studied variables". Do you think you cannot make a solid conclusion with these results? I hope that my comments will help you to improve the manuscript. I did not find any structural problems of methodology, which would lead to rejecting it; but it needs to be carefully reviewed.

Author Response

I first would like to thank you for taking the time to review our publication and providing feedback. Your suggestions greatly helped to improve the manuscript and for that, I am very grateful. Here is my response to each of your suggestions and comments:

1 and 2:  We added information to the short summary as well as the introduction to make the objectives of the research project clearer. We also added information to make the gap in knowledge clearer and added a statement about the gap in knowledge. We also added an objective statement to each of the aforementioned sections. We also added more information to help strengthen the technical justification and reorganized the introduction section. 

3: Upon your suggestion we decide to omit the word supplementation as the cranberry vine was being incorporated into the grain itself.

4: Additional details were provided to help clarify the experimental design 

5: We opted to separate the results and discussion sections based on your recommendation. This has greatly helped the flow of the publication. It also allowed us to make certain parts clearer and more concise while adding additional information to other parts of the discussion. 

6. Clear conclusions were added to the conclusions section 

Reviewer 2 Report

This study generates information on the effects of a non-traditional food supplement on the productive characteristics of ewes and lambs. The information is novel and complete, however major revision should be done in presentation, writing and English style before its approval. 

It is requested to improve the English style, in particular to edit the manuscript to a more technical English.

The development of the Introduction, Simple Summary and Summary do not lead to the objective. It is suggested to review their content. It describes the importance of using the cranberry vine supplement in terms of parasite control, but then proposes another objective and measures other variables.

 It is suggested to present Results and Discussion in separate sections.

Materials and methods: Correct the titles and footnotes of figures and tables. The titles must detail all the content of the table. Some acronyms at the bottom of the tables contain errors.

The Discussion could be shorter and more concise.

Author Response

I would first like to thank you for taking the time out of your busy schedule to review our manuscript. Your constructive suggestions greatly helped us improve the quality of the manuscript and for that, I am greatly appreciative.

Here is the line-by-line responses to your suggestions:

1: We read the manuscript over and removed all informal language and tried to improve the flow of the manuscript.

2. We added a clear objective statement to both the simple summary and introduction. We also elaborated on some key points to make it clear that we are trying to determine the safety and efficacy of CVP based feeds in sheep during pregnancy and lactation. 

3. We separated the results and discussion section and this helped out the quality of the manuscript greatly!

4. Footnotes were fixed and titles of the tables were improved to accurately represent what they are presenting

5. The discussion section was able to be condensed when we separated it out per your suggestion. 

Author Response

I would first like to thank you for taking the time out of your day to review this manuscript. Your constructive and thorough review of this manuscript help us to improve it and for that I am greatly appreciative.

Here is a point-by-point response of the improvements we made based on your suggestions/feedback:

1) We added information about the power analyses that we performed prior to the study to ensure that we had an adequate number of animals for statistical significance. 

2) We added information about how the ewes were housed and what was considered to be the experimental unit (the ewe and not the pen). We also provided information about any normalization that we made to account for ewes giving birth throughout the season (and not on the same day). We tried to the best of our abilities to balance the anticipated number of lambs (twin vs singleton) within each treatment group and this was stated

3) We separated out the results and discussion section which greatly helped the quality of the manuscript. 

We would like to thank you for the thorough line-by-line edits that you presented to us for the minor revisions. I know that these take additional time to do, and we appreciate your extra efforts. We went through and fixed the typos that you found as well as the redundant statements and abbreviations that need to either be added or spelled out. 

Round 2

Reviewer 1 Report

Congratulations, the text had a giant scientific evolution. I was satisfied with the text in its present format; so I recommend the publication.